# Critical initialisation for deep signal propagation in noisy rectifier neural networks

**Arnu Pretorius**[*]
Computer Science Division
CAIR[†]
Stellenbosch University

**Elan Van Biljon**
Computer Science Division
Stellenbosch University

**Steve Kroon**
Computer Science Division
Stellenbosch University

**Herman Kamper**
Department of Electrical and Electronic Engineering
Stellenbosch University

## Abstract

Stochastic regularisation is an important weapon in the arsenal of a deep learning practitioner. However, despite recent theoretical advances, our understanding of how noise influences signal propagation in deep neural networks remains limited. By extending recent work based on mean field theory, we develop a new framework for signal propagation in stochastic regularised neural networks. Our *noisy signal propagation* theory can incorporate several common noise distributions, including additive and multiplicative Gaussian noise as well as dropout. We use this framework to investigate initialisation strategies for noisy ReLU networks. We show that no critical initialisation strategy exists using additive noise, with signal propagation exploding regardless of the selected noise distribution. For multiplicative noise (e.g. dropout), we identify alternative critical initialisation strategies that depend on the second moment of the noise distribution. Simulations and experiments on real-world data confirm that our proposed initialisation is able to stably propagate signals in deep networks, while using an initialisation disregarding noise fails to do so. Furthermore, we analyse correlation dynamics between inputs. Stronger noise regularisation is shown to reduce the depth to which discriminatory information about the inputs to a noisy ReLU network is able to propagate, even when initialised at criticality. We support our theoretical predictions for these trainable depths with simulations, as well as with experiments on MNIST and CIFAR-10.[‡]

## 1 Introduction

Over the last few years, advances in network design strategies have made it easier to train large networks and have helped to reduce overfitting. These advances include improved weight initialisation strategies (Glorot and Bengio, 2010; Saxe et al., 2014; Sussillo and Abbott, 2014; He et al., 2015; Mishkin and Matas, 2016), non-saturating activation functions (Glorot et al., 2011) and stochastic regularisation techniques (Srivastava et al., 2014). Authors have noted, for instance, the critical dependence of successful training on noise-based methods such as dropout (Krizhevsky et al., 2012; Dahl et al., 2013).

---

[*]Correspondence: arnupretorius@gmail.com

[†]CSIR/SU Centre for Artificial Intelligence Research.

[‡]Code to reproduce all the results is available at https://github.com/ElanVB/noisy_signal_prop

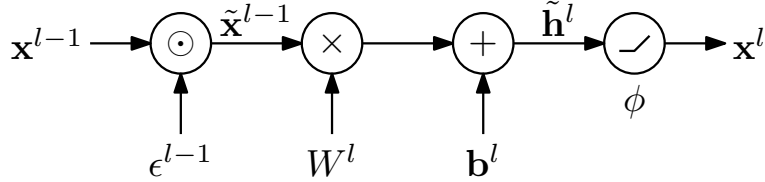

Figure 1: *Noisy layer recursion.* The input $\mathbf{x}^{l-1}$ from the previous layer gets corrupted by the sampled noise $\epsilon^{l-1}$, either by vector addition or component-wise multiplication, producing the noisy inputs $\tilde{\mathbf{x}}^{l-1}$. The $l^{th}$ layer's corrupted pre-activations are then computed by multiplication with the layer weight matrix $W^l$, followed by a vector addition of the biases $\mathbf{b}^l$. Finally, the inputs to the next layer are simply the activations of the current layer, *i.e.* $\mathbf{x}^l = \phi(\tilde{\mathbf{h}}^l)$.

In many cases, successful results arise only from effective combination of these advances. Despite this interdependence, our theoretical understanding of how these mechanisms and their interactions affect neural networks remains impoverished.

One approach to studying these effects is through the lens of deep neural signal propagation. By modelling the empirical input variance dynamics at the point of random initialisation, Saxe et al. (2014) were able to derive equations capable of describing how signal propagates in nonlinear fully connected feed-forward neural networks. This "mean field" theory was subsequently extended by Poole et al. (2016) and Schoenholz et al. (2017), in particular, to analyse signal correlation dynamics. These analyses highlighted the existence of a critical boundary at initialisation, referred to as the "edge of chaos". This boundary defines a transition between ordered (vanishing), and chaotic (exploding) regimes for neural signal propagation. Subsequently, the mean field approximation to random neural networks has been employed to analyse other popular neural architectures (Yang and Schoenholz, 2017; Xiao et al., 2018; Chen et al., 2018).

This paper focuses on the effect of noise on signal propagation in deep neural networks. Firstly we ask: How is signal propagation in deep neural networks affected by noise? To gain some insight into this question, we extend the mean field theory developed by Schoenholz et al. (2017) for the special case of dropout noise, into a generalised framework capable of describing the signal propagation behaviour of stochastically regularised neural networks for different noise distributions.

Secondly we ask: How much are current weight initialisation strategies affected by noise-induced regularisation in terms of their ability to initialise at a critical point for stable signal propagation? Using our derived theory, we investigate this question specifically for rectified linear unit (ReLU) networks. In particular, we show that no such critical initialisation exists for arbitrary zero-mean additive noise distributions. However, for multiplicative noise, such an initialisation is shown to be possible, given that it takes into account the amount of noise being injected into the network. Using these insights, we derive novel critical initialisation strategies for several different multiplicative noise distributions.

Finally, we ask: Given that a network is initialised at criticality, in what way does noise influence the network's ability to propagate useful information about its inputs? By analysing the correlation between inputs as a function of depth in random deep ReLU networks, we highlight the following: even though the statistics of individual inputs are able to propagate arbitrarily deep at criticality, *discriminatory information* about the inputs becomes lost at shallower depths as the noise in the network is increased. This is because in the later layers of a random noisy network, the internal representations from different inputs become uniformly correlated. Therefore, the application of noise regularisation directly limits the trainable depth of critically initialised ReLU networks.

## 2 Noisy signal propagation

We begin by presenting mean field equations for stochastically regularised fully connected feed-forward neural networks, allowing us to study noisy signal propagation for a variety of noise distributions. To understand how noise influences signal propagation in a random network given an input $\mathbf{x}^0 \in \mathbb{R}^{D_0}$, we inject noise into the model

$$\tilde{\mathbf{h}}^l = W^l(\mathbf{x}^{l-1} \odot \epsilon^{l-1}) + \mathbf{b}^l, \qquad \text{for } l = 1, ..., L \tag{1}$$

using the operator $\odot$ to denote either addition or multiplication where $\epsilon^l$ is an input noise vector, sampled from a pre-specified noise distribution. For additive noise, the distribution is assumed to be zero mean, for multiplicative noise distributions, the mean is assumed to be equal to one. The weights $W^l \in \mathbb{R}^{D_l \times D_{l-1}}$ and biases $\mathbf{b}^l \in \mathbb{R}^{D_l}$ are sampled i.i.d. from zero mean Gaussian distributions with variances $\sigma_w^2/D_{l-1}$ and $\sigma_b^2$, respectively, where $D_l$ denotes the dimensionality of the $l^{th}$ hidden layer in the network. The hidden layer activations $\mathbf{x}^l = \phi(\tilde{\mathbf{h}}^l)$ are computed element-wise using an activation function $\phi(\cdot)$, for layers $l = 1, ..., L$. Figure 1 illustrates this recursive sequence of operations.

To describe forward signal propagation for the model in (1), we make use of the mean field approximation as in Poole et al. (2016) and analyse the statistics of the internal representations of the network in expectation over the parameters and the noise. Since the weights and biases are sampled from zero mean Gaussian distributions with pre-specified variances, we can approximate the distribution of the pre-activations at layer $l$, in the large width limit, by a zero mean Gaussian with variance

$$\tilde{q}^l = \sigma_w^2 \left\{ \mathbb{E}_z \left[ \phi \left( \sqrt{\tilde{q}^{l-1}} z \right)^2 \right] \odot \mu_2^{l-1} \right\} + \sigma_b^2, \tag{2}$$

where $z \sim \mathcal{N}(0, 1)$ (see Section A.1 in the supplementary material). Here, $\mu_2^l = \mathbb{E}_\epsilon[(\epsilon^l)^2]$ is the second moment of the noise distribution being sampled from at layer $l$. The initial input variance is given by $q^0 = \frac{1}{D_0} \mathbf{x}^0 \cdot \mathbf{x}^0$. Furthermore, to study the behaviour of a pair of signals from two different inputs, $\mathbf{x}^{0,a}$ and $\mathbf{x}^{0,b}$, passing through the network, we can compute the covariance at each layer as

$$\tilde{q}_{ab}^l = \sigma_w^2 \mathbb{E}_{z_1} \left[ \mathbb{E}_{z_2} \left[ \phi(\tilde{u}_1)\phi(\tilde{u}_2) \right] \right] + \sigma_b^2 \tag{3}$$

where $\tilde{u}_1 = \sqrt{\tilde{q}_{aa}^{l-1}} z_1$ and $\tilde{u}_2 = \sqrt{\tilde{q}_{bb}^{l-1}} \left[ \tilde{c}^{l-1} z_1 + \sqrt{1 - (\tilde{c}^{l-1})^2} z_2 \right]$, with the correlation between inputs at layer $l$ given by $\tilde{c}^l = \tilde{q}_{ab}^l / \sqrt{\tilde{q}_{aa}^l \tilde{q}_{bb}^l}$. Here, $q_{aa}^l$ is the variance of $\tilde{\mathbf{h}}_j^{l,a}$ (see Section A.2 in the supplementary material for more details).

For the backward pass, we use the equations derived in Schoenholz et al. (2017) to describe error signal propagation.[1] In the context of mean field theory, the expected magnitude of the gradient at each layer can be shown to be proportional to the variance of the error, $\tilde{\delta}_i^l = \phi'(\tilde{\mathbf{h}}_i^l) \sum_{j=1}^{D_{l+1}} \tilde{\delta}_j^{l+1} W_{ji}^{l+1}$. This allows for the distribution of the error signal at layer $l$ to be approximated by a zero mean Gaussian with variance

$$\tilde{q}_\delta^l = \tilde{q}_\delta^{l+1} \frac{D_{l+1}}{D_l} \sigma_w^2 \mathbb{E}_z \left[ \phi' \left( \sqrt{\tilde{q}^l} z \right)^2 \right]. \tag{4}$$

Similarly, for noise regularised networks, the covariance between error signals can be shown to be

$$\tilde{q}_{ab,\delta}^l = \tilde{q}_{ab,\delta}^{l+1} \frac{D_{l+1}}{D_{l+2}} \sigma_w^2 \mathbb{E}_{z_1} \left[ \mathbb{E}_{z_2} \left[ \phi'(\tilde{u}_1)\phi'(\tilde{u}_2) \right] \right], \tag{5}$$

where $\tilde{u}_1$ and $\tilde{u}_2$ are defined as was done in the forward pass.

Equations (2)-(5) fully capture the relevant statistics that govern signal propagation for a random network during both the forward and the backward pass. In the remainder of this paper, we consider, as was done by Schoenholz et al. (2017), the following necessary condition for training: "for a random network to be trained information about the inputs should be able to propagate forward through the network, and information about the gradients should be able to propagate backwards through the network." The behaviour of the network at this stage depends on the choice of activation, noise regulariser and initial parameters. In the following section, we will focus on networks that use the Rectified Linear Unit (ReLU) as activation function. The chosen noise regulariser is considered a design choice left to the practitioner. Therefore, whether a random noisy ReLU network satisfies the above stated necessary condition for training largely depends on the starting parameter values of the network, *i.e.* its initialisation.

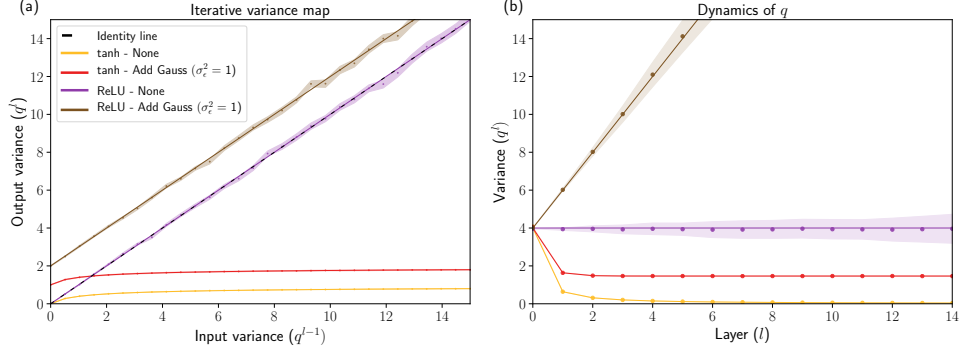

Figure 2: *Deep signal propagation with and without noise*. **(a)**: Iterative variance map. **(b)**: Variance dynamics during forward signal propagation. In (a) and (b), lines correspond to theoretical predictions and points to numerical simulations (means over 50 runs with shaded one standard deviation bounds), for noiseless tanh (yellow) and noiseless ReLU (purple) networks, as well as for noisy tanh (red) and noisy ReLU (brown) networks regularised using additive noise from a standard Gaussian. Both tanh networks use $(\sigma_w, \sigma_b) = (1, 0)$, the "Xavier" initialisation (Glorot and Bengio, 2010), while the ReLU networks use $(\sigma_w, \sigma_b) = (\sqrt{2}, 0)$ the "He" initialisation (He et al., 2015). In our experiments, we use network layers consisting of 1000 hidden units (see Section C in the supplementary material for more details on all our simulated experiments).

## 3   Critical initialisation for noisy rectifier networks

Unlike the tanh nonlinearity investigated in previous work (Poole et al., 2016; Schoenholz et al., 2017), rectifying activation functions such as ReLU are unbounded. This means that the statistics of signal propagation through the network is not guaranteed to naturally stabilise through saturating activations, as shown in Figure 2.

A point on the identity line in Figure 2 (a) represents a fixed point to the recursive variance map in equation (2). At a fixed point, signal will stably propagate through the remaining layers of the network. For tanh networks, such a fixed point always exists irrespective of the initialisation, or the amount of noise injected into the network. For ReLU networks, this is not the case. Consider the "He" initialisation (He et al., 2015) for ReLU, commonly used in practice. In (b), we plot the variance dynamics for this initialisation in purple and observe stable behaviour. But what happens when we inject noise into each network? In the case of tanh (shown in red), the added noise simply shifts the fixed point to a new stable value. However, for ReLU, the noise entirely destroys the fixed point for the "He" initialisation, making signal propagation unstable. This can be seen in (a), where the variance map for noisy ReLU (shown in brown) moves off the identity line entirely, causing the signal in (b) to explode.

Therefore, to investigate whether signal can stably propagate through a random *noisy* ReLU network, we examine (2) more closely, which for ReLU becomes (see Section B.1 in supplementary material)

$$\tilde{q}^l = \sigma_w^2 \left[ \frac{\tilde{q}^{l-1}}{2} \odot \mu_2 \right] + \sigma_b^2. \tag{6}$$

For ease of exposition we assume equal noise levels at each layer, *i.e.* $\mu_2^l = \mu_2, \forall l$. A critical initialisation for a noisy ReLU network occurs when the tuple $(\sigma_w, \sigma_b, \mu_2)$ provides a fixed point $\tilde{q}^*$, to the recurrence in (6). This at least ensures that the statistics of individual inputs to the network will be preserved throughout the first forward pass. The existence of such a solution depends on the type of noise that is injected into the network. In the case of additive noise, $\tilde{q}^* = \sigma_w^2 \frac{1}{2} \tilde{q}^* + \mu_2 \sigma_w^2 + \sigma_b^2$, implying that the only critical point initialisation for non-zero $\tilde{q}^*$ is given by $(\sigma_w, \sigma_b, \mu_2) = (\sqrt{2}, 0, 0)$. Therefore, critical initialisation is not possible using any amount of zero-mean additive noise, regardless of the noise distribution. For multiplicative noise, $\tilde{q}^* = \sigma_w^2 \frac{1}{2} \tilde{q}^* \mu_2 + \sigma_b^2$, so the solution $(\sigma_w, \sigma_b, \mu_2) = \left( \sqrt{\frac{2}{\mu_2}}, 0, \mu_2 \right)$ provides a critical initialisation for noise distributions with mean one and a non-zero second moment $\mu_2$. For example, in the case of multiplicative Gaussian noise, $\mu_2 = \sigma_\epsilon^2 + 1$, yielding critical initialisation with $(\sigma_w, \sigma_b) = \left( \sqrt{\frac{2}{\sigma^2 + 1}}, 0 \right)$. For dropout noise,

Table 1: Critical point initialisation for noisy ReLU networks.

| DISTRIBUTION | P($\epsilon$) | $\mu_2$ | CRITICAL INITIALISATION |
|---|---|---|---|
| — ADDITIVE NOISE — | | | |
| GAUSSIAN | $\mathcal{N}(0, \sigma_\epsilon^2)$ | $\sigma_\epsilon^2$ | $(\sigma_w, \sigma_b, \sigma_\epsilon) = (\sqrt{2}, 0, 0)$ |
| LAPLACE | $Lap(0, \beta)$ | $2\beta^2$ | $(\sigma_w, \sigma_b, \beta) = (\sqrt{2}, 0, 0)$ |
| — MULTIPLICATIVE NOISE — | | | |
| GAUSSIAN | $\mathcal{N}(1, \sigma_\epsilon^2)$ | $(\sigma_\epsilon^2 + 1)$ | $(\sigma_w, \sigma_b, \sigma_\epsilon) = \left(\sqrt{\frac{2}{\sigma_\epsilon^2 + 1}}, 0, \sigma_\epsilon\right)$ |
| LAPLACE | $Lap(1, \beta)$ | $(2\beta^2 + 1)$ | $(\sigma_w, \sigma_b, \beta) = \left(\sqrt{\frac{2}{2\beta^2 + 1}}, 0, \beta\right)$ |
| POISSON | $Poi(1)$ | $2$ | $(\sigma_w, \sigma_b, \lambda) = (1, 0, 1)$ |
| DROPOUT | $\mathrm{P}(\epsilon = \frac{1}{p}) = p,$ $\mathrm{P}(\epsilon = 0) = 1 - p$ | $\frac{1}{p}$ | $(\sigma_w, \sigma_b, p) = (\sqrt{2p}, 0, p)$ |

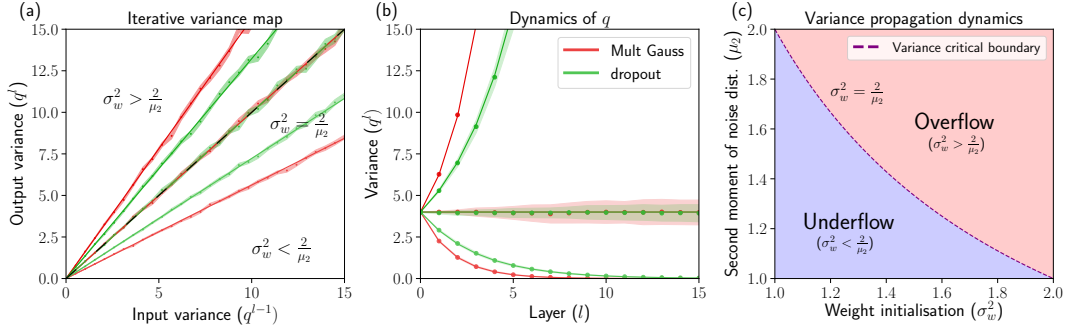

Figure 3: *Critical initialisation for noisy ReLU networks.* **(a)**: Iterative variance map. **(b)**: Variance dynamics during forward signal propagation. In (a) and (b), lines correspond to theoretical predictions and points to numerical simulations. Dropout ($p = 0.6$) is shown in green for different initialisations, $\sigma_w^2 = 2(0.6) = \frac{2}{\mu_2}$ (critical), $\sigma_w^2 = (1.15)^2 \frac{2}{(0.6)^{-1}} > \frac{2}{\mu_2}$ (exploding signal) and $\sigma_w^2 = (0.85)^2 \frac{2}{(0.6)^{-1}} < \frac{2}{\mu_2}$ (vanishing signal). Similarly, multiplicative Gaussian noise ($\sigma_\epsilon = 0.25$) is shown in red with $\sigma_w^2 = \frac{2}{(0.25)^2 + 1} = \frac{2}{\mu_2}$ (critical), $\sigma_w^2 = (1.25)^2 \frac{2}{\mu_2}$ (exploding) and $\sigma_w^2 = (0.75)^2 \frac{2}{\mu_2}$ ( vanishing). **(c)**: Variance critical boundary for initialisation, separating numerical overflow and underflow signal propagation regimes.

$\mu_2 = 1/p$ (with $p$ the probability of retaining a neuron); thus, to initialise at criticality, we must set $(\sigma_w, \sigma_b) = (\sqrt{2p}, 0)$. Table 1 summarises critical initialisations for some commonly used noise distributions. We also note that similar results can be derived for other rectifying activation functions; for example, for multiplicative noise the critical initialisation for parametric ReLU (PReLU) activations (with slope parameter $\alpha$) is given by $(\sigma_w, \sigma_b, \mu_2) = \left(\sqrt{\frac{2}{\mu_2(\alpha^2 + 1)}}, 0, \mu_2\right)$.

To see the effect of initialising on or off the critical point for ReLU networks, Figure 3 compares the predicted versus simulated variance dynamics for different initialisation schemes. For schemes not initialising at criticality, the variance map in (a) no longer lies on the identity line and as a result the forward propagating signal in (b) either explodes, or vanishes. In contrast, the initialisations derived above lie on the critical boundary between these two extremes, as shown in (c) as a function of the noise. By compensating for the amount of injected noise, the signal corresponding to the initialisation $\sigma_w^2 = \frac{2}{\mu_2}$ is preserved in (b) throughout the entire forward pass, with roughly constant variance dynamics.

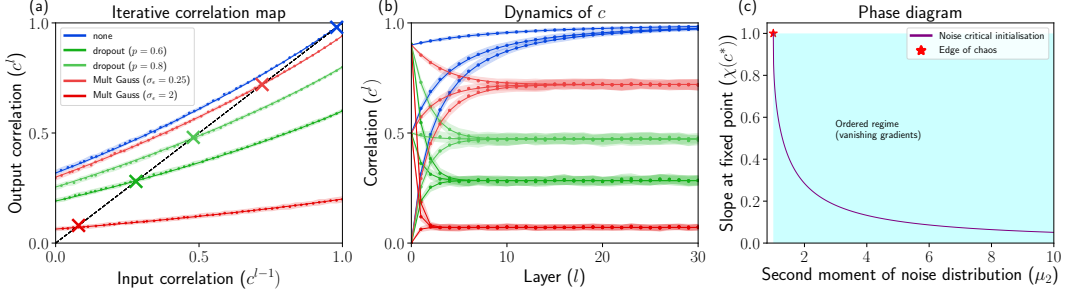

Figure 4: *Propagating correlation information in noisy ReLU networks.* **(a)**: Iterative correlation map with fixed points indicated by "X" marks on the identity line. **(b)**: Correlation dynamics during forward signal propagation. In (a) and (b), lines correspond to theoretical predictions and points to numerical simulations. All simulated networks were initialised at criticality for each noise type and level. **(c)**: Slope at the fixed point correlation as a function of the amount of noise injected into the network.

Next, we investigate the correlation dynamics between inputs. Assuming that (6) is at its fixed point $\tilde{q}^*$, which exists only if $\sigma_w^2 = \frac{2}{\mu_2}$, the correlation map for a noisy ReLU network is given by (see Section B.2 in supplementary material)

$$\tilde{c}^l = \frac{1}{\mu_2} \left\{ \frac{\tilde{c}^{l-1}\sin^{-1}\left(\tilde{c}^{l-1}\right) + \sqrt{1 - (\tilde{c}^{l-1})^2}}{\pi} + \frac{\tilde{c}^{l-1}}{2} \right\}. \tag{7}$$

Figure 4 plots this theoretical correlation map against simulated dynamics for different noise types and levels. For no noise, the fixed point $c^*$ in (a) is situated at one (marked with an "X" on the blue line). The slope of the blue line indicates a non-decreasing function of the input correlations. After a certain depth, inputs end up perfectly correlated irrespective of their starting correlation, as shown in (b). In other words, random deep ReLU networks lose discriminatory information about their inputs as the depth of the network increases, even when initialised at criticality. When noise is added to the network, inputs decorrelate and $c^*$ moves away from one. However, more importantly, correlation information in the inputs become lost at shallower depths as the noise level increases, as can be seen in (b).

How quickly a random network loses information about its inputs depends on the rate of convergence to the fixed point $c^*$. Using this observation, Schoenholz et al. (2017) derived so-called depth scales $\xi_c$, by assuming $|c^l - c^*| \sim e^{-l/\xi_c}$. These scales essentially control the feasible depth at which networks can be considered trainable, since they may still allow useful correlation information to propagate through the network. In our case, the depth scale for a noisy ReLU network under this assumption can be shown to be (see Section B.3 in supplementary material)

$$\xi_c = -1/\ln\left[\chi(c^*)\right], \tag{8}$$

where

$$\chi(c^*) = \frac{1}{\mu_2\pi} \left[\sin^{-1}\left(c^*\right) + \frac{\pi}{2}\right]. \tag{9}$$

The exponential rate assumption underlying the derivation of (8) is supported in Figure 5, where for different noise types and levels, we plot $|c^l - c^*|$ as a function of depth on a log-scale, with corresponding linear fits (see panels (a) and (c)). We then compare the theoretical depth scales from (8) to actual depth scales obtained through simulation (panels (b) and (d)), as a function of noise and observe a good fit for non-zero noise levels.[4] We thus find that noise limits the depth at which critically initialised ReLU networks are expected to perform well through training.

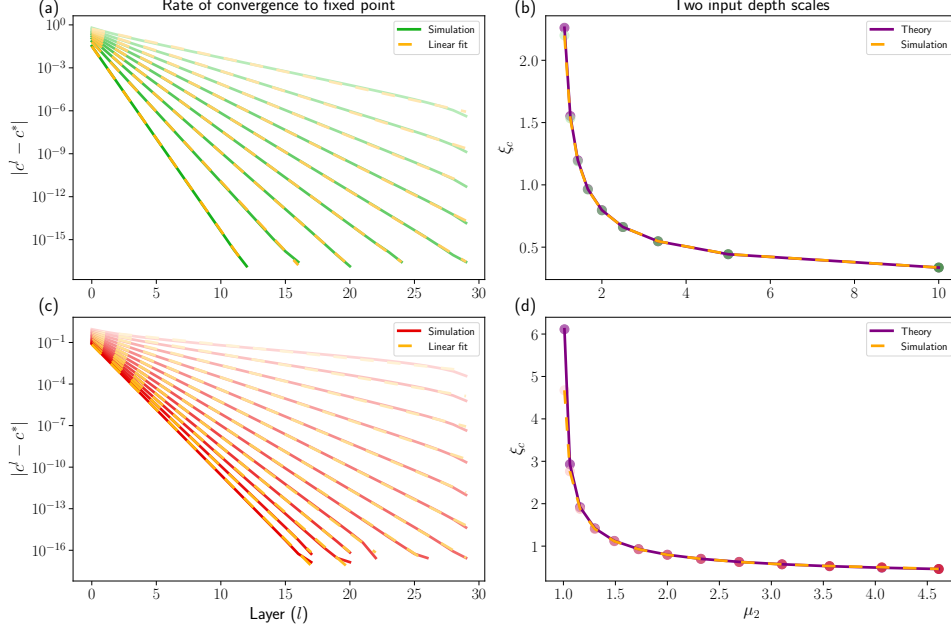

Figure 5: *Noise dependent depth scales for training.* **(a)**: Linear fits (dashed lines) to $|c^l - c^*|$ as a function of depth on a log-scale (solid lines) for varying amounts of dropout ($p = 0.1$ to $p = 0.9$ by $0.1$). **(b)**: Theoretical depth scales (solid lines) versus empirically inferred scales (dashed lines) per dropout rate. Scales are inferred noting that if $|c^l - c^*| \sim e^{-l/\xi_c}$, then a linear fit, $al + b$, in the logarithmic domain gives $\xi_c \approx -\frac{1}{a}$, for large $l$. In other words, the negative inverse slope of a linear fit to the log differences in correlation should match the theoretical values for $\xi_c$. Therefore, we compare $\xi_c = -1/\ln\left[\chi(c^*)\right]$ to $-\frac{1}{a}$ for different levels of noise. **(c)** - **(d)**: Similar to (a) and (b), but for Gaussian noise ($\sigma_\epsilon = 0.1$ to $\sigma_\epsilon = 1.9$ by $0.15$).

We next briefly discuss error signal propagation during the backward pass for noise regularised ReLU networks. When critically initialised, the error variance recurrence relation in (4) for these networks is (see Section B.4 in supplementary material)

$$\tilde{q}_\delta^l = \tilde{q}_\delta^{l+1} \frac{D_{l+1}}{D_l \mu_2}, \qquad (10)$$

with the covariance between error signals in (5), given by (see Section B.5 in supplementary material)

$$\tilde{q}_{ab,\delta}^l = \tilde{q}_{ab,\delta}^{l+1} \frac{D_{l+1}}{D_{l+2}} \chi(c^*). \qquad (11)$$

Note the explicit dependence on the width of the layers of the network in (10) and (11). We first consider constant width networks, where $D_{l+1} = D_l$, for all $l = 1, ..., L$. For any amount of multiplicative noise, $\mu_2 > 1$, and we see from (10) that gradients will tend to vanish for large depths. Furthermore, Figure 4 (c) plots $\chi(c^*)$ as a function of $\mu_2$. As $\mu_2$ increases from one, $\chi(c^*)$ decreases from one. Therefore, from (11), we also find that error signals from different inputs will tend to decorrelate at large depths.

Interestingly, for non-constant width networks, stable gradient information propagation may still be possible. If the network architecture adapts to the amount of noise being injected by having the widths of the layers grow as $D_{l+1} = D_l \mu_2$, then (10) should be at its fixed point solution. For example, in the case of dropout $D_{l+1} = D_l/p$, which implies that for any $p < 1$, each successive layer in the network needs to grow in width by a factor of $1/p$ to promote stable gradient flow. Similarly, for multiplicative Gaussian noise, $D_{l+1} = D_l(\sigma_\epsilon^2 + 1)$, which requires the network to grow in width unless $\sigma_\epsilon^2 = 0$. Similarly, if $D_{l+2} = D_{l+1}\chi(c^*) = D_l \mu_2 \chi(c^*)$ in (11), the covariance of the error signal should be preserved during the backward pass, for arbitrary values of $\mu_2$ and $\chi(c^*)$.

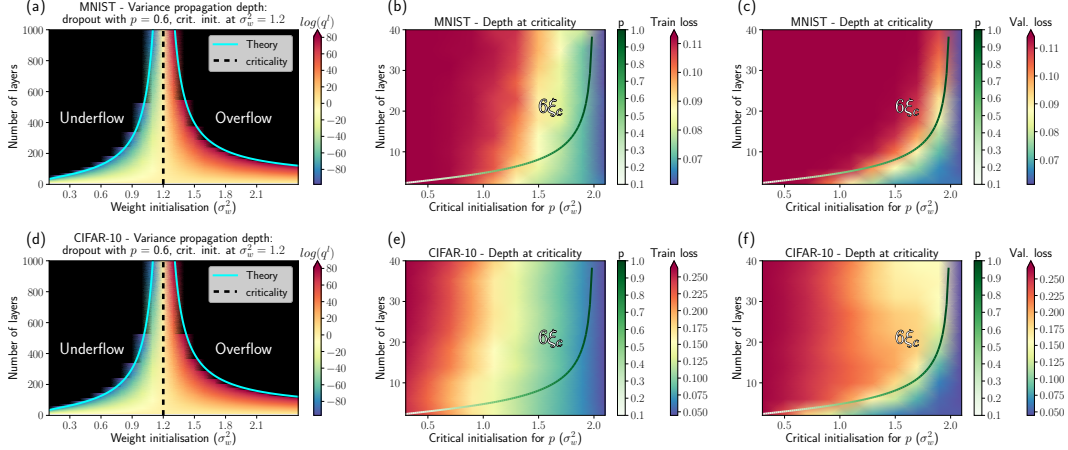

Figure 6: *Depth scale experiments on MNIST and CIFAR-10.* **(a)** Variance propagation dynamics for MNIST on and off the critical point initialisation (dashed black line) with dropout ($p = 0.6$). The cyan curve represents the theoretical boundary at which numerical instability issues are predicted to occur and is computed as $L^* = \ln(K)/\ln(\frac{\sigma_w^2}{2}\mu_2)$, where $K$ is the largest (or smallest) positive number representable by the computer. Specifically, we use 32-bit floating point numbers and set $K = 3.4028235 \times 10^{38}$, if $\sigma_w^2 > \frac{2}{\mu_2}$ and $K = 1.1754944 \times 10^{-38}$, if $\sigma_w^2 < \frac{2}{\mu_2}$. **(b)** Depth scales fit to the training loss on MNIST for networks initialised at criticality for dropout rates $p = 0.1$ (severe dropout) to $p = 1$ (no dropout). **(c)** Depth scales fit to the validation loss on MNIST. **(d) - (f)**: Similar to (a) - (c), but for CIFAR-10. For each plot we highlight trends by smoothing the colour grid (for non smoothed versions see Section C.5 in the supplementary material).

## 4 Experimental results

From our analysis of deep noisy ReLU networks in the previous section, we expect that a necessary condition for such a network to be trainable, is that the network be initialised at criticality. However, whether the layer widths are varied or not for the sake of backpropagation, the correlation dynamics in the forward pass may still limit the depth at which these networks perform well.

We therefore investigate the performance of noise-regularised deep ReLU networks on real-world data. First, we validate the derived critical initialisation. As the depth of the network increases, any initialisation strategy that does not factor in the effects of noise, will cause the forward propagating signal to become increasingly unstable. For very deep networks, this might cause the signal to either explode or vanish, even within the first forward pass, making the network untrainable. To test this, we sent inputs from MNIST and CIFAR-10 through ReLU networks using dropout (with $p = 0.6$) at varying depths and for different initialisations of the network. Figure 6 (a) and (d) shows the evolution of the input statistics as the input propagates through each network for the different data sets. For initialisations not at criticality, the variance grows or shrinks rapidly to the point of causing numerical overflow or underflow (indicated by black regions). For deep networks, this can happen well before any signal is able to reach the output layer. In contrast, initialising at criticality (as shown by the dashed black line), allows for the signal to propagate reliably even at very large depths. Furthermore, given the floating point precision, if $\sigma_w^2 \neq \frac{2}{\mu_2}$, we can predict the depth at which numerical overflow (or underflow) will occur by solving for $L^*$ in $K = \left(\sigma_w^2 \mu_2/2\right)^{L^*} q^0$, where $K$ is the largest (or smallest) positive number representable by the computer (see Section C.4 in supplementary material). These predictions are shown by the cyan line and provide a good fit to the empirical limiting depth from numerical instability.

We now turn to the issue of limited trainability. Due to the loss of correlation information between inputs as a function of noise and network depth, we expect noisy ReLU networks not to be able to perform well beyond certain depths. We investigated depth scales for ReLU networks with dropout initialised at criticality: we trained 100 networks on MNIST and CIFAR-10 for 200 epochs using SGD and a learning rate of $10^{-3}$ with dropout rates ranging from 0.1 to 1 for varying depths. The results

are shown in Figure 6 (see Section C.5 of the supplementary material for additional experimental results). For each network configuration and noise level, the critical initialisation $\sigma_w^2 = \frac{2}{\mu_2}$ was used. We indeed observe a relationship between depth and noise on the loss of a network, even at criticality. Interestingly, the line $6\xi_c$ (Schoenholz et al., 2017), seems to track the depth beyond which the relative performance on the validation loss becomes poor, more so than on the training loss. However, in both cases, we find that even modest amounts of noise can limit performance.

## 5  Discussion

By developing a general framework to study signal propagation in noisy neural networks, we were able to show how different stochastic regularisation strategies may impact the flow of information in a deep network. Focusing specifically on ReLU networks, we derived novel critical initialisation strategies for multiplicative noise distributions and showed that no such critical initialisations exist for commonly used additive noise distributions. At criticality however, our theory predicts that the statistics of the input should remain within a stable range during the forward pass and enable reliable signal propagation for noise regularised deep ReLU networks. We verified these predictions by comparing them with numerical simulations as well as experiments on MNIST and CIFAR-10 using dropout and found good agreement.

Interestingly, we note that a dropout rate of $p = 0.5$ has often been found to work well for ReLU networks (Srivastava et al., 2014). The critical initialisation corresponding to this rate is $(\sigma_w, \sigma_b) = (\sqrt{2p}, 0) = (1, 0)$. This is exactly the "Xavier" initialisation proposed by Glorot and Bengio (2010), which prior to the development of the "He" initialisation, was often used in combination with dropout (Simonyan and Zisserman, 2014). This could therefore help to explain the initial success associated with this specific dropout rate. Similarly, Srivastava et al. (2014) reported that adding multiplicative Gaussian noise where $\epsilon \sim \mathcal{N}(1, \sigma_\epsilon^2)$, with $\sigma_\epsilon^2 = 1$, also seemed to perform well, for which the critical initialisation is $\left(\sqrt{\frac{2}{\sigma_\epsilon^2+1}}, 0\right) = (1, 0)$, again corresponding to the "Xavier" method.

Although our initialisations ensure that individual input statistics are preserved, we further analysed the correlation dynamics between inputs and found the following: at large depths inputs become predictably correlated with each other based on the amount of noise injected into the network. As a consequence, the representations for different inputs to a deep network may become indistinguishable from each other in the later layers of the network. This can make training infeasible for noisy ReLU networks of a certain depth and depends on the amount of noise regularisation being applied.

We now note the following shortcomings of our work: firstly, our findings only apply to fully connected feed-forward neural networks and focus almost exclusively on the ReLU activation function. Furthermore, we limit the scope of our architectural design to a recursive application of a dense layer followed by a noise layer, whereas in practice a larger mix of layers is usually required to solve a specific task.

Ultimately, we are interested in reducing the number of decisions that need to made when designing deep neural networks and understanding the implications of those decisions on network behaviour and performance. Any machine learning engineer exploring a neural network based solution to a practical problem will be faced with a large number of possible design decisions. All these decisions cost valuable time to explore. In this work, we hope to have at least provided some guidance in this regard, specifically when choosing between different initialisation strategies for noise regularised ReLU networks and understanding their associated implications.

## Acknowledgements

We would like to thank the reviewers for their insightful comments which improved the quality of this work. Furthermore, we would like to thank Google, the CSIR/SU Centre for Artificial Intelligence Research (CAIR) as well as the Science Faculty and the Postgraduate and International Office of Stellenbosch University for financial support. Finally, we gratefully acknowledge the support of NVIDIA Corporation with the donation of a Titan Xp GPU used for this research.

## Footnotes

[1]It is, however, important to note that the derivation relies on the assumption that the weights used in the forward pass are sampled independently from those used during backpropagation.

[4]We note Hayou et al. (2018) recently showed that the rate of convergence for noiseless ReLU networks is not exponential, but polynomial instead. Interestingly, keeping with the exponential rate assumption, we indeed find that the discrepancy between our theoretical depth scales from (8) and our simulated depth scales, is largest at very low noise levels. However, at more typical noise levels, such as a dropout rate of $p = 0.5$ for example, the assumption seems to provide a close fit, with good agreement between theory and simulation.

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
