[Supplementary Material · supplementary.pdf]

## Supplementary Material

In this section, we provide additional details of derivations and experimental results presented in the paper.

## A  Signal propagation in noise regularised neural networks

To review, given an input $\mathbf{x}^0 \in \mathbb{R}^{D_0}$, we consider the following noisy random network model

$$\tilde{\mathbf{h}}^l = W^l(\mathbf{x}^{l-1} \odot \epsilon^{l-1}) + \mathbf{b}^l, \qquad \text{for } l = 1, ..., L \qquad (12)$$

where we inject noise into the model using the operator $\odot$ to denote either addition or multiplication. The vector $\epsilon^l$ is an input noise vector, sampled from a pre-specified noise distribution. For additive noise, the distribution is assumed to be zero mean. Whereas for multiplicative noise distributions, the mean is assumed to be equal to one. The weights $W^l \in \mathbb{R}^{D_l \times D_{l-1}}$ and biases $\mathbf{b}^l \in \mathbb{R}^{D_l}$ are sampled i.i.d. from zero mean Gaussian distributions with variances $\sigma_w^2/D_{l-1}$ and $\sigma_b^2$, respectively, where $D_l$ denotes the dimensionality of the $l^{th}$ hidden layer in the network. The hidden layer activations $\mathbf{x}^l = \phi(\tilde{\mathbf{h}}^l)$ are computed element-wise using an activation function $\phi(\cdot)$, for layers $l = 1, ..., L$.

### A.1  Single input signal propagation

We consider the network's behavior at initialisation. In this setting, the expected mean (over the weights, biases and noise distribution) of a unit in the pre-activations $\tilde{\mathbf{h}}_j^l$ for a single signal passing through the network will be zero with variance

$$\begin{aligned}
\tilde{q}^l &= \mathbb{E}_{\mathbf{w},\mathbf{b},\epsilon}[(\tilde{\mathbf{h}}_j^l)^2] \\
&= \mathbb{E}_{\mathbf{w},\epsilon}[\{\mathbf{w}^{l,j} \cdot (\mathbf{x}_j^{l-1} \odot \epsilon_j^{l-1})\}^2] + \mathbb{E}_{\mathbf{b}}[(\mathbf{b}_j^l)^2] \\
&= \sigma_w^2 \frac{1}{D_{l-1}} \sum_{j=1}^{D_{l-1}} \left[ \phi(\tilde{\mathbf{h}}_j^{l-1})^2 \odot \mathbb{E}_\epsilon[(\epsilon_j^{l-1})^2] \right] + \sigma_b^2 \ ,
\end{aligned}$$

where we use $\mathbf{w}^{l,j}$ to denote the $j$-th row of $W^l$. The second last line relies on the bias distribution being zero mean, while the final step makes use of the independence between the inputs and the noise in the multiplicative case, and the noise being zero mean in the additive case. Furthermore, to ensure the expected value of the pre-activations remain unbiased, we only consider additive noise distributions with zero mean and multiplicative noise distributions with a mean equal to one. As in Poole et al. (2016), we make the self averaging assumption and consider the large layer width case where the previous layer's pre-activations are assumed to be Gaussian with zero mean and variance $\tilde{q}^{l-1}$. This gives the following noisy variance map

$$\tilde{q}^l = \sigma_w^2 \left\{ \mathbb{E}_z \left[ \phi \left( \sqrt{\tilde{q}^{l-1}} z \right)^2 \right] \odot \mu_2^{l-1} \right\} + \sigma_b^2, \qquad (13)$$

where $z \sim \mathcal{N}(0, 1)$ and $\mu_2^l = \mathbb{E}_\epsilon[(\epsilon^l)^2]$ is the second moment of the noise distribution being sampled from at layer $l$. The initial input variance is given by $q^0 = \frac{1}{D_0}\mathbf{x}^0 \cdot \mathbf{x}^0$.

### A.2  Two input signal propagation

To study the behaviour of a pair of signals, $\mathbf{x}^{0,a}$ and $\mathbf{x}^{0,b}$, passing through the network, we can compute the covariance in expectation over the noise and the parameters as

$$\begin{aligned}
\tilde{q}_{ab}^l &= \mathbb{E}_{\mathbf{w},\mathbf{b},\epsilon}[\tilde{\mathbf{h}}_j^{l,a}\tilde{\mathbf{h}}_j^{l,b}] \\
&= \mathbb{E}_{\mathbf{w},\mathbf{b},\epsilon}\left[ \left( \mathbf{w}^{l,j} \cdot (\mathbf{x}_j^{l-1,a} \odot \epsilon_j^{l-1,a}) + \mathbf{b}_j^l \right) \left( \mathbf{w}^{l,j} \cdot (\mathbf{x}_j^{l-1,b} \odot \epsilon_j^{l-1,b}) + \mathbf{b}_j^l \right) \right] \\
&= \mathbb{E}_{\mathbf{w},\mathbf{b},\epsilon}\left[ \left( \mathbf{w}^{l,j} \cdot (\mathbf{x}_j^{l-1,a} \odot \epsilon_j^{l-1,a}) \right) \left( \mathbf{w}^{l,j} \cdot (\mathbf{x}_j^{l-1,b} \odot \epsilon_j^{l-1,b}) \right) \right] \\
&\quad + \mathbb{E}_{\mathbf{w},\mathbf{b},\epsilon}\left[ \left( \mathbf{w}^{l,j} \cdot (\mathbf{x}_j^{l-1,a} \odot \epsilon_j^{l-1,a}) \right) \mathbf{b}_j^l \right] \\
&\quad + \mathbb{E}_{\mathbf{w},\mathbf{b},\epsilon}\left[ \left( \mathbf{w}^{l,j} \cdot (\mathbf{x}_j^{l-1,b} \odot \epsilon_j^{l-1,b}) \right) \mathbf{b}_j^l \right] \\
&\quad + \mathbb{E}_{\mathbf{w},\mathbf{b},\epsilon}\left[ (\mathbf{b}_j^l)^2 \right].
\end{aligned}$$

Since the noise is i.i.d and we have that $\mathbb{E}_{\mathbf{b}}[b_j^l] = 0$, we find that

$$\tilde{q}_{ab}^l = \mathbb{E}_{\mathbf{w}}\left[\left(\mathbf{w}^{l,j} \cdot \mathbf{x}_j^{l-1,a}\right)\left(\mathbf{w}^{l,j} \cdot \mathbf{x}_j^{l-1,b}\right)\right] + \mathbb{E}_{\mathbf{b}}\left[(b_j^l)^2\right] \tag{14}$$

$$= \sigma_w^2 \frac{1}{D_{l-1}} \sum_{j=1}^{D_{l-1}} \left[\phi\left(\tilde{\mathbf{h}}_j^{l-1,a}\right)\phi\left(\tilde{\mathbf{h}}_j^{l-1,b}\right)\right] + \sigma_b^2, \tag{15}$$

which in the large width limit becomes

$$\tilde{q}_{ab}^l = \sigma_w^2 \mathbb{E}_{z_1}\left[\mathbb{E}_{z_2}\left[\phi(\tilde{u}_1)\phi(\tilde{u}_2)\right]\right] + \sigma_b^2 \tag{16}$$

where $\tilde{u}_1 = \sqrt{\tilde{q}_{aa}^{l-1}} z_1$ and $\tilde{u}_2 = \sqrt{\tilde{q}_{bb}^{l-1}}\left[\tilde{c}^{l-1}z_1 + \sqrt{1 - (\tilde{c}^{l-1})^2}z_2\right]$, with the correlation between inputs at layer $l$ given by

$$\tilde{c}^l = \tilde{q}_{ab}^l / \sqrt{\tilde{q}_{aa}^l \tilde{q}_{bb}^l}. \tag{17}$$

Here, $z_i \sim \mathcal{N}(0,1)$ for $i = 1, 2$ and $q_{aa}^l$ is the variance of $\tilde{\mathbf{h}}_j^{l;a}$.

## B  Signal propagation in noise regularised ReLU networks

In this section, we give additional details of theoretical results presented in the paper that were specifically derived for noisy ReLU networks.

### B.1  Variance of input signals

Let $f(z) = \frac{e^{-z^2/2}}{\sqrt{2\pi}}$, then the variance map in (13) using ReLU, i.e. $\phi(a) = \max(0, a)$, becomes

$$\tilde{q}^l = \sigma_w^2 \left[\int_{-\infty}^{\infty} f(z)\phi\left(\sqrt{\tilde{q}^{l-1}}z\right)^2 dz\right] \odot \mu_2 + \sigma_b^2$$

$$= \sigma_w^2 \left[\int_{-\infty}^{0} f(z)\phi\left(\sqrt{\tilde{q}^{l-1}}z\right)^2 dz + \int_{0}^{\infty} f(z)\phi\left(\sqrt{\tilde{q}^{l-1}}z\right)^2 dz\right] \odot \mu_2 + \sigma_b^2$$

$$= \sigma_w^2 \left[\tilde{q}^{l-1}\int_{0}^{\infty} f(z)z^2 dz\right] \odot \mu_2 + \sigma_b^2$$

$$= \sigma_w^2 \left[\frac{\tilde{q}^{l-1}}{2} \odot \mu_2\right] + \sigma_b^2. \tag{18}$$

### B.2  Correlation between input signals

Assuming that the variance map in (18) is at its fixed point $\tilde{q}^*$, which exits only if $\sigma_w^2 = \frac{2}{\mu_2}$, the correlation map in (16) for a noisy ReLU network is given by

$$\tilde{c}^l = \frac{2}{\mu_2 \tilde{q}^*} \int_{-\infty}^{\infty} \int_{-\infty}^{\infty} f(z_1)f(z_2)\phi(\tilde{u}_1)\phi(\tilde{u}_2)dz_2 dz_1 + \sigma_b^2 \tag{19}$$

where $\phi(a) = \max(a, 0)$, $f(z_i) = \frac{e^{-z_i^2/2}}{\sqrt{2\pi}}$, $\tilde{u}_1 = \sqrt{\tilde{q}^*}z_1$ and $\tilde{u}_2 = \sqrt{\tilde{q}^*}\left[\tilde{c}^{l-1}z_1 + \sqrt{1-(\tilde{c}^{l-1})^2}z_2\right]$. Note that

$$\tilde{u}_1 \begin{cases} \geq 0, \text{if } z_1 > 0 \\ < 0, \text{Otherwise} \end{cases}$$

$$\tilde{u}_2 \begin{cases} \geq 0, \text{if } z_2 > \frac{-\tilde{c}^{l-1}z_1}{\sqrt{1-(\tilde{c}^{l-1})^2}} \\ < 0, \text{Otherwise} \end{cases} ,$$

therefore (19) becomes

$$
\begin{aligned}
\tilde{c}^l &= \frac{2}{\mu_2 \tilde{q}^*} \int_0^\infty \int_{\frac{-\tilde{c}^{l-1} z_1}{\sqrt{1-(\tilde{c}^{l-1})^2}}}^\infty f(z_1) f(z_2) \tilde{u}_1 \tilde{u}_2 dz_2 dz_1 + \sigma_b^2 \\
&= \frac{2}{\mu_2 \tilde{q}^*} \sigma_w^2 \int_0^\infty \int_{\frac{-\tilde{c}^{l-1} z_1}{\sqrt{1-(\tilde{c}^{l-1})^2}}}^\infty f(z_1) f(z_2) \sqrt{\tilde{q}^*} z_1 \sqrt{\tilde{q}^*} \left[ \tilde{c}^{l-1} z_1 + \sqrt{1-(\tilde{c}^{l-1})^2} z_2 \right] dz_2 dz_1 + \sigma_b^2 \\
&= \frac{2\tilde{c}^{l-1}}{\mu_2} \int_0^\infty \int_{\frac{-\tilde{c}^{l-1} z_1}{\sqrt{1-(\tilde{c}^{l-1})^2}}}^\infty f(z_1) f(z_2) z_1^2 dz_2 dz_1 \\
&\qquad\qquad + \frac{2\sqrt{1-(\tilde{c}^{l-1})^2}}{\mu_2} \int_0^\infty \int_{\frac{-\tilde{c}^{l-1} z_1}{\sqrt{1-(\tilde{c}^{l-1})^2}}}^\infty f(z_1) f(z_2) z_1 z_2 dz_2 dz_1. \qquad (20)
\end{aligned}
$$

The first term in (20) can then be written as

$$
\frac{2\tilde{c}^{l-1}}{\mu_2} \left\{ \int_0^\infty \int_{\frac{-\tilde{c}^{l-1} z_1}{\sqrt{1-(\tilde{c}^{l-1})^2}}}^0 f(z_1) f(z_2) z_1^2 dz_2 dz_1 + \int_0^\infty \int_0^\infty f(z_1) f(z_2) z_1^2 dz_2 dz_1 \right\}. \qquad (21)
$$

In (21), the first term inside the braces is given by

$$
\begin{aligned}
\int_0^\infty \int_{\frac{-\tilde{c}^{l-1} z_1}{\sqrt{1-(\tilde{c}^{l-1})^2}}}^0 f(z_1) f(z_2) z_1^2 dz_2 dz_1 &= \frac{1}{2} \int_0^\infty f(z_1) z_1^2 \text{erf} \left( \frac{\tilde{c}^{l-1} z_1}{\sqrt{1-(\tilde{c}^{l-1})}} \right) dz_1 \\
&= \frac{1}{2\pi} \left[ \tilde{c}^{l-1} \sqrt{1-(\tilde{c}^{l-1})^2} + \tan^{-1} \left( \frac{\tilde{c}^{l-1}}{\sqrt{1-(\tilde{c}^{l-1})^2}} \right) \right] \\
&= \frac{1}{2\pi} \left[ \tilde{c}^{l-1} \sqrt{1-(\tilde{c}^{l-1})^2} + \sin^{-1} \left( \tilde{c}^{l-1} \right) \right] \qquad (22)
\end{aligned}
$$

with $\text{erf}(a) = \frac{1}{\pi} \int_{-a}^a e^{-t^2} dt$. The second term inside the braces in (21) equals

$$
\begin{aligned}
\int_0^\infty \int_0^\infty f(z_1) f(z_2) z_1^2 dz_2 dz_1 &= \frac{1}{2} \int_0^\infty f(z_1) z_1^2 dz_1 \\
&= \frac{1}{4}. \qquad (23)
\end{aligned}
$$

Therfore, (21) becomes

$$
\frac{(\tilde{c}^{l-1})^2}{\mu_2 \pi} \sqrt{1-(\tilde{c}^{l-1})^2} + \frac{\tilde{c}^{l-1}}{\mu_2 \pi} \sin^{-1} \left( \tilde{c}^{l-1} \right) + \frac{\tilde{c}^{l-1}}{2\mu_2} \qquad (24)
$$

Similarly, the second term in (20) can be split up as follows

$$
\frac{2\sqrt{1-(\tilde{c}^{l-1})^2}}{\mu_2} \left\{ \int_0^\infty \int_{\frac{-\tilde{c}^{l-1} z_1}{\sqrt{1-(\tilde{c}^{l-1})^2}}}^0 f(z_1) f(z_2) z_1 z_2 dz_2 dz_1 + \int_0^\infty \int_0^\infty f(z_1) f(z_2) z_1 z_2 dz_2 dz_1 \right\}. \qquad (25)
$$

The first term inside the braces of (25) is

$$
\begin{aligned}
\int_0^\infty \int_{\frac{-\tilde{c}^{l-1} z_1}{\sqrt{1-(\tilde{c}^{l-1})^2}}}^0 f(z_1) f(z_2) z_1 z_2 dz_2 dz_1 &= \frac{1}{\sqrt{2\pi}} \int_0^\infty f(z_1) z_1 \left[ e^{-\frac{\tilde{c}^{l-1} z_1^2}{2(1-(\tilde{c}^{l-1})^2)}} - 1 \right] dz_1 \\
&= \frac{1}{\sqrt{2\pi}} \left\{ \frac{1-(\tilde{c}^{l-1})^2}{\sqrt{2\pi}} - \frac{1}{\sqrt{2\pi}} \right\} \\
&= -\frac{(\tilde{c}^{l-1})^2}{2\pi} \qquad (26)
\end{aligned}
$$

and the second term is

$$\int_0^\infty \int_0^\infty f(z_1)f(z_2)z_1z_2dz_2dz_1 = \frac{1}{\sqrt{2\pi}} \int_0^\infty f(z_1)z_1dz_1$$

$$= \frac{1}{2\pi}. \tag{27}$$

Putting these two terms together, (25) becomes

$$-\frac{(\tilde{c}^{l-1})^2}{\mu_2\pi}\sqrt{1-(\tilde{c}^{l-1})^2} + \frac{1}{\mu_2\pi}\sqrt{1-(\tilde{c}^{l-1})^2}. \tag{28}$$

Finally, summing all the terms in (24) and (28) gives (19) as

$$\tilde{c}^l = \frac{1}{\mu_2}\left\{ \frac{\tilde{c}^{l-1}\sin^{-1}\left(\tilde{c}^{l-1}\right) + \sqrt{1-(\tilde{c}^{l-1})^2}}{\pi} + \frac{\tilde{c}^{l-1}}{2}\right\}. \tag{29}$$

We note that for the noiseless case, (29) is identical to the result recently obtained by Hayou et al. (2018), where the authors used a slightly different approach.

## B.3 Depth scales for trainability

We recap the result in Schoenholz et al. (2017) and adapt the derivation for the specific case of a noisy ReLU network. Let $c^l = c^* + \varepsilon^l$, such that as long as $\lim_{l\to\infty} c^l = c^*$ exist we have that $\varepsilon \to 0$ as $l \to \infty$. Then Schoenholz et al. (2017) derived the following asymptotic recurrence relation

$$\varepsilon^{l+1} = \varepsilon^l \chi(c^*) + \mathcal{O}((\varepsilon^l)^2), \tag{30}$$

where

$$\chi(c^*) = \sigma_w^2 \mathbb{E}_{z_1}\left[\mathbb{E}_{z_2}\left[\phi'(\tilde{u}_1^*)\phi'(\tilde{u}_2^*)\right]\right], \tag{31}$$

with $\tilde{u}_1^* = \tilde{u}_1 = \sqrt{\tilde{q}^*}z_1$ and $\tilde{u}_2^* = \sqrt{\tilde{q}^*}\left[\tilde{c}^* z_1 + \sqrt{1-(\tilde{c}^*)^2}z_2\right]$. Now, specifically for a noisy ReLU network where $\sigma_w^2 = \frac{2}{\mu_2}$, we have that

$$\chi(c^*) = \frac{2}{\mu_2}\int_{-\infty}^\infty \int_{-\infty}^\infty f(z_1)f(z_2)\phi'(\tilde{u}_1^*)\phi'(\tilde{u}_2^*)dz_2dz_1$$

$$= \frac{2}{\mu_2}\int_0^\infty \int_{-\frac{c^*z_1}{\sqrt{1-(c^*)^2}}}^\infty f(z_1)f(z_2)dz_2dz_1$$

$$= \frac{2}{\mu_2}\int_0^\infty f(z_1)\frac{1}{2}\left[\mathrm{erf}\left(\frac{c^*z_1}{\sqrt{2}\sqrt{1-(c^*)^2}}\right) + 1\right]dz_1$$

$$= \frac{2}{\mu_2}\left[\frac{1}{2\pi}\tan^{-1}\left(\frac{c^*}{\sqrt{1-(c^*)^2}}\right) + \frac{1}{4}\right]$$

$$= \frac{1}{\mu_2\pi}\left[\sin^{-1}\left(c^*\right) + \frac{\pi}{2}\right] \tag{32}$$

Note that $\chi(c^*)$ is a constant, thus for large $l$ the solution to the recurrence relation in (30) is expected to be exponential, *i.e.* $\varepsilon^l \sim e^{-l/\xi_c}$. Here $\xi_c$, is considered the *depth scale*, which controls how deep discriminatory information about the inputs can propagate through the network. We can then solve for $\xi_c$ to find

$$\xi_c = -1/\ln(\chi(c^*)) = -\ln\left[\frac{\sin^{-1}\left(c^*\right)}{\mu_2\pi} + \frac{1}{2\mu_2}\right]^{-1}. \tag{33}$$

## B.4 Variance of error signals

Under the mean field assumption, Schoenholz et al. (2017) approximates the error signal at layer $l$ by a zero mean Gaussian with variance

$$\tilde{q}_\delta^l = \tilde{q}_\delta^{l+1} \frac{D_{l+1}}{D_l} \sigma_w^2 \mathbb{E}_z \left[ \phi' \left( \sqrt{\tilde{q}^l} z \right)^2 \right], \tag{34}$$

where $\tilde{q}_\delta^l = \mathbb{E}[(\tilde{\delta}_i^l)^2]$, with $\tilde{\delta}_i^l = \phi'(\tilde{\mathbf{h}}_i^l) \sum_{j=1}^{D_{l+1}} \tilde{\delta}_j^{l+1} W_{ji}^{l+1}$. In our context, for a critically initialised noisy ReLU network we have that

$$\tilde{q}_\delta^l = \tilde{q}_\delta^{l+1} \frac{D_{l+1}}{D_l} \frac{2}{\mu_2} \int_0^\infty f(z) dz \tag{35}$$

$$= \tilde{q}_\delta^{l+1} \frac{D_{l+1}}{D_l} \frac{1}{\mu_2}. \tag{36}$$

## B.5 Correlation between error signals

The covariance between error signals is approximated using

$$\tilde{q}_{ab,\delta}^l = \tilde{q}_{ab,\delta}^{l+1} \frac{D_{l+1}}{D_{l+2}} \sigma_w^2 \mathbb{E}_{z_1} \left[ \mathbb{E}_{z_2} \left[ \phi'(\tilde{u}_1) \phi'(\tilde{u}_2) \right] \right], \tag{37}$$

where $\tilde{u}_1$ and $\tilde{u}_2$ are defined as was done in the forward pass. Here, we simply use the result in (32) for noisy ReLU networks to find

$$\tilde{q}_{ab,\delta}^l = \tilde{q}_{ab,\delta}^{l+1} \frac{D_{l+1}}{D_{l+2}} \chi(c^*) \tag{38}$$

$$= \tilde{q}_{ab,\delta}^{l+1} \frac{D_{l+1} \left[ \sin^{-1}(c^*) + \frac{\pi}{2} \right]}{D_{l+2} \mu_2 \pi}. \tag{39}$$

## C Experimental details

In this section we provide additional details regarding our experiments in the paper. Code to reproduce all the experiments is available at https://github.com/ElanVB/noisy_signal_prop.

### C.1 Input data

For all experiments the network input data properties that remain consistent (unless stated otherwise) are as follows: each observation consists of 1000 features and each feature value is drawn i.i.d. from a standard normal distribution.

### C.2 Variance propagation dynamics

The experiments conducted to gather results for Figures 2 and 3 aim to empirically show the relationship between the variances at arbitrary layers in a neural network.

*Iterative map*: For the results depicted in Figures 2 (a) and 3 (a), the experimental set up is as follows. The data used as input to these experiments comprises of 30 sets of 30 observations. The input is scaled such that the variance of observations within each set is the same and the variance across each set is different and forms a range of $q_{\text{set}} \in [0, 15]$. As such, our results are averaged over 30 observations and 50 samplings of initial weights to a single hidden-layer network.

*Convergence dynamics*: For the results depicted in Figures 2 (b) and 3 (b), the experimental set up is as follows. The data used as input to these experiments comprises of a set of 50 observations scaled such that each observation's variance is four ($q = 4$). As such, our results are averaged over 50 observations and 50 samplings of initial weights to a 15 hidden-layer network.

### C.3 Correlation propagation dynamics

The experiments conducted to gather results for Figure 4 and 5 aim to empirically show the relationship between the correlations of observations at arbitrary layers in a neural network.

*Iterative map*: For the results depicted in Figure 4 (a), the experimental set up is as follows. The data used as input to these experiments comprises of 50 sets of 50 observations. The first observation in each set is sampled from a standard normal distribution and subsequent observations are generated such that the correlation between the first element and the $i^{\text{th}}$ element form a range of $\text{corr}_{0,i} \in [0, 1]$. As such, our results are averaged over 50 observations and 50 samplings of initial weights to a single hidden-layer network.

*Convergence dynamics*: For the results depicted in Figure 4 (b), the experimental set up is as follows. The data used as input to these experiments comprises of three sets of 50 equally correlated observations. Each set has a different correlation value such that $\text{corr}_{\text{set}} \in \{0, 0.5, 0.9\}$. As such, our results are averaged over 50 observations and 50 samplings of initial weights to a 15 hidden-layer network.

*Confirmation of exponential rate of convergence for correlations*: This section discusses how the results depicted in Figure 5 are acquired. These experiments support the assumption that the rate of convergence for correlations is exponential when using noise regularisation with rectifier neural networks. The experimental set up for this section is very similar to that of the above convergence dynamics experiment, the only difference being the statistics we calculate from the correlation values. The aspect of this experiment that may seem the most unclear is the reason why we claim that the negative inverse slope of a linear fit to the $\log$ differences in correlation should match the theoretical values for $\xi_c$. The derivation to justify this is as follows. If a good fit of the form $al + b$ can be found in the logarithmic domain for the rate of convergence, it would strongly indicate that the convergence rate is exponential. Following this, we set the problem up like so:

$$|c^l - c^*| \approx e^{-l/\xi_c}$$
$$\therefore \ln\left(|c^l - c^*|\right) \approx \frac{-l}{\xi_c}.$$

Let us now assume that $\ln\left(|c^l - c^*|\right)$ can be linearly approximated:

$$\therefore \ln\left(|c^l - c^*|\right) \approx al + b,$$
$$\therefore al + b \approx \frac{-l}{\xi_c},$$
$$\therefore \xi_c \approx \frac{-l}{al + b}.$$

Since we are concerned with deep neural networks, we can take the limit as $l$ becomes arbitrarily large and see that as $l$ grows the effect of $b$ decreases ($\lim_{l\to\infty} |al| \gg |b|$). Thus, we continue like so:

$$\lim_{l\to\infty} \xi_c \approx \lim_{l\to\infty} \frac{-l}{al}$$
$$\approx -\frac{1}{a}.$$

Thus, we have come to the finding that if the correlation rate of convergence is exponential and we work with deep neural networks, the negative inverse slope of a linear fit to the $\log$ differences in correlation should match the theoretical values for $\xi_c$. Figure 5 shows that the theory closely matches this approximation.

## C.4 Depth scales

This section handles the experiments conducted related to determining the maximum depth variance information can stably propagate through a network and the depth at which these networks can be trained, both depicted in Figure 6.

The MNIST and CIFAR-10 datasets were used and were pre-processed using standard techniques. Throughout these experiments mini-batches of 128 observations were used.

*Variance depth scales*: The experiments depicted in Figures 6 (a) and (d) are interested in testing the numerical stability of networks initialised using different $\sigma_w^2$ values while using 32-bit floating point

numbers. To test the depth of stable variance propagation, a network with 1000 hidden layers is used. The network used in this experiment makes use of dropout with $p = 0.6$, where $p$ is the probability of keeping a neuron's value, thus the critical value for $\sigma_w^2$ is 1.2. As such, a linearly spaced range of $\sigma_w^2 \in [0.1, 2.5]$ is used to select 25 different values.

We use the following approach to predict the depth beyond which variances become numerically unstable. At criticality for multiplicative noise $(\sigma_w, \sigma_b) = (\sqrt{2/\mu_2}, 0)$, however, for weights initialised off this critical point (18) becomes

$$
\begin{aligned}
\tilde{q}^l &= \tilde{q}^{l-1} \left( \frac{\sigma_w^2 \mu_2}{2} \right) \\
&= \left[ \tilde{q}^{l-2} \left( \frac{\sigma_w^2 \mu_2}{2} \right) \right] \left( \frac{\sigma_w^2 \mu_2}{2} \right) \\
&= \tilde{q}^0 \left( \frac{\sigma_w^2 \mu_2}{2} \right)^l .
\end{aligned}
\tag{40}
$$

If $\sigma_w^2 > \frac{2}{\mu_2}$, we let $\tilde{q}^l = K$, where $K$ is the largest positive number representable by the computer. In our case, using 32-bit floating point precision, this number is equal to $3.4028235 \times 10^{38}$. Otherwise, if $\sigma_w^2 < \frac{2}{\mu_2}$ we select $K = 1.1754944 \times 10^{-38}$, the smallest possible positive number. Furthermore, let $L^*$ represent the layer $l$ in (40) at which the value $K$ is reached, then we can scale our input data such that $\tilde{q}^0 = 1$ and solve for $L^*$ to find

$$
L^* = \ln(K) / \ln \left( \frac{\sigma_w^2 \mu_2}{2} \right) .
\tag{41}
$$

Therefore, we expect numerical instability issues to occur beyond a depth of $L^*$.

*Trainable depth scales*: The experiments depicted in Figures 6 (b), (c), (e) and (f) are concerned with determining at what depth a critically initialised network with a specified dropout rate can train effectively. To this end, 10 linearly spaced values for dropout on the range $p \in [0.1, 1.0]$ and 10 linearly spaced network depths on the integer range $l \in [2, 40]$ are tested.

The task presented to the network in this experiment is to learn the identity function within 200 epochs. As such, the network is set up as an auto-encoder and uses stochastic gradient decent with a learning rate of $10^{-3}$. The input data is divided into a training and validation set, each containing 50000 and 10000 observations respectively.

## C.5  Additional results

In this section we provide some additional experiments on the training dynamics of deep noisy ReLU networks from different initialisations.

In Figure 7 we compare the standard "He" initialisation (blue) with the critical initialisation (green) for a ReLU network with dropout regularisation ($p = 0.8$). By not initialising at criticality due to dropout noise, the variance map for the "He" strategy no longer lies on the identity line in (a) and as a result, the forward propagating signal can be seen to explode in (b). However, by compensating for the amount of injected noise, the above derived critical initialisation for dropout preserves the signal throughout the entire forward pass, with roughly constant variance dynamics.

Next, we provide some additional experiments on the trainability of deep ReLU networks with dropout on real-world data sets.

From our analysis in the paper, we expect that as the depth of the network increases, any initialisation strategy that does not factor in the effects of noise, will cause the forward propagating signal to become increasingly unstable. For very deep networks, this might cause the signal to either explode or vanish, even within the first forward pass, making the network untrainable.

To test this, we trained a denoising autoencoder network with dropout noise ($p = 0.6$) on MNIST and CIFAR-10 using squared reconstruction loss. We consider several network depths ($L = 30, 100, 200$), learning rates ($\alpha = 0.1, 0.01, 0.001, 0.0001$) and optimisation procedures (SGD and Adam), with 1000 neurons in each layer. The results for training on CIFAR-10 are shown in Figure 8 for both the "He" intialisation (blue) and the critical dropout initialisation (green). (For MNIST, see Figure 9; the

Figure 7: *Critical initialisation for ReLU networks with dropout.* Lines correspond to theoretical predictions and points to numerical simulations, for random ReLU networks with dropout ($p = 0.8$), initialised according to the method proposed by He et al. (2015) (blue) and at criticality (green). **(a)**: Iterative variance map where the identity line is displayed as a dashed black line. **(b)**: Variance dynamics during forward signal propagation.

Figure 8: Comparing the "He" initialisation strategy to critical dropout initialisation for ReLU networks using dropout ($p = 0.6$) on CIFAR-10. While networks initialised at criticality (green) are able to train at large depths ($L = 200$) as seen in the bottom row, networks initialised with the "He" strategy (blue) become untrainable irrespective of the chosen learning rate or optimisation procedure. An "X" marks the point at which a network completely stopped training. Training losses and number of network updates are shown in log-scale.

core trends and resulting conclusions regarding network trainability is the same for both data sets, which we discuss below.)

As the depth increases, moving from the top to the bottom row in Figure 8, networks initialised at the critical point for dropout seem to remain trainable even up to a depth of 200 layers (we see the loss start to decrease over five epochs). In contrast, networks using the "He" initialisation become increasingly more difficult to train, with no training taking place at very large depths. These findings make sense in terms of the variance dynamics analysed in the paper, however, these experimental successes seem to run counter to our theoretical analysis of trainable depth scales (this contradiction can also be seen in Figure 6). Understanding this discrepancy is of particular interest to us.

To verify that the lack of training in Figure 8 is due to poor signal propagation, we plot the empirical variance of the pre-activations in Figure 10, for the first forward pass of a 200 layer autoencoder

Figure 9: Comparing the "He" initialisation strategy to critical dropout initialisation for ReLU networks using dropout ($p = 0.6$) on MNIST. While networks initialised at criticality (green) are able to train at large depths ($L = 200$) as seen in the bottom row, networks initialised with the "He" strategy (blue) become untrainable irrespective of the chosen learning rate or optimisation procedure. An "X" marks the point at which a network completely stopped training. Training losses and number of network updates are shown in log-scale.

Figure 10: *Variance dynamics for signal propagation in the first forward pass for a 200 layer autoencoder network fed a batch of 500 training examples from CIFAR-10.* **(a)** Exploding activation variance (blue) reaching overflow levels (marked with a red "X") for the "He" intialisation, with no signal reaching the output layer (shown in log-scale). **(b)** Zoomed in display of the roughly constant variance dynamics in (a) for the critical dropout initialisation.

network. For the "He" initialisation, the variance in (a) grows rapidly to the point of causing numerical instability and overflow (indicated by the red dashed line), well before any signal is able to reach the output layer. However as shown in (b), by initialising at criticality, signal is able to propagate reliably even at large depths.

Figure 11: *Depth scale experiments on MNIST and CIFAR-10.* **(a)** Depth scales fit to the training loss on MNIST for networks initialised at criticality for dropout rates $p = 0.1$ (severe dropout) to $p = 1$ (no dropout). **(b)** Depth scales fit to the validation loss on MNIST. **(c) - (d)**: Similar to (a) - (c), but for CIFAR-10.