[Reviews · NeurIPS 2018]

Reviewer 1



The paper proposed a new framework for initializing deep neural networks. The proposed initialization is based on the analysis on variances of layer outputs when the inputs contains additive or multiplicative noises. The idea of designing initialization heuristics to maintain the scale of outputs stable over different layers isn’t something very novel. The “He” initialization methods mentioned in the paper was originally proposed to maintain the standard deviation of each layer. This paper extended this idea to the general additive and multiplicative noises by deriving the explicit formula for the variances of layer output. The idea was interesting and it leads to some neat result on the initialization approach for network with dropout noise. But I feel there’s still a big gap between the analysis the paper and the practical phenomenon the paper tries to address. The paper only discussed the distribution of the layer output when the weights are random, but in realty the weights are no longer randomly distributed after some iterations of back-propagation. So the analysis in the paper cannot really explain why the new initialization work in practice. Even though the authors provided some discussion about that in the final section, it’s worthy some full-fledged discussion if the authors really want to develop a theory around the proposed initialization approach. It’s also kind of hard to choose which initialization methods can be used in practice as we can never know the exact noise type from the input distribution. So I can hardly see any practical implication of the analysis proposed in the paper.

Reviewer 2



This paper analyzes signal propagation in vanilla fully-connected neural networks in the presence of noise. For ReLU networks, it concludes that the initial weight variance should be adjusted from the "He" initialization to account for the noise scale. Various empirical simulations corroborate this claim. Generally speaking, I believe studying signal propagation in random neural networks is a powerful way to build better initialization schemes and examining signal propagation in the presence of noise is an interesting direction. The paper is well-written and easy to read. However, I found the analysis and conclusions of this work to be rather weak and I do not believe it is appropriate for publication at NIPS at this time. The technical contributions of this work are readily derived from eqn. (5), which is a fairly simple modification of the recursion relation studied in [1,2]. From eqn. (5), it is easy to see that multiplicative noise can be absorbed into \sigma_w, and additive noise can be absorbed into \sigma_b. While this is a useful observation, to me it seems fairly trivial from the technical perspective and insufficient to serve as the backbone of the entire paper. Perhaps more importantly, the paper focuses entirely on "q", i.e. pre-activation variances, and does not analyze the propagation of "c", i.e. pre-activation correlations. The latter is necessary for understanding the phase diagram, the stability of fixed points, and the trainable depth scales. Even if the phase diagram ends up looking identical to the noise-free case (modulo redefinitions of \sigma_w and \sigma_b to account for the noise), this point should have been discussed and investigated. The trainability experiments in Figure 4 go out to depth 200. While this is certainly quite deep, it may not be deep enough to demonstrate the breakdown of trainability arising from ill-conditioned Jacobians. Indeed, in [3], critically-initialized networks with ill-conditioned Jacobians can train out to depth 200, albeit much more slowly than their well-conditioned counterparts. Overall, while I like the general idea behind this paper, I do not find the results significant enough to merit publication at NIPS at this stage. [1] Poole, Ben, et al. "Exponential expressivity in deep neural networks through transient chaos." Advances in neural information processing systems. 2016. [2] Schoenholz, Samuel S., et al. "Deep Information Propagation." ICLR 2017. [3] Pennington, Jeffrey, Samuel Schoenholz, and Surya Ganguli. "Resurrecting the sigmoid in deep learning through dynamical isometry: theory and practice." Advances in neural information processing systems. 2017. Edit: After reviewing the authors' response and examining the extensive new experiments, I have increased my score to 7 and vote for acceptance. The authors added new analysis on the phase diagram, convergence rates of correlations, and many other important metrics. I believe these analyses will significantly improve the paper. I still think the technical meat of the paper is rather weak, but given the extensive new experiments I think the paper is worthy of publication.

Reviewer 3



The paper proposes an initialization for noisy rectifier neural network.For multiplicative noise the paper shows that a similar analysis to Poole et al can be carried to obtain recursive equations for infinite number of hidden layers, then by solving the fixed point equation one finds the correct initialization. for additive noise the paper shows that the fixed point equation does not have a solution. The paper has nice insights on better initialization when using dropout noise and draws nice comparisons and connections to Xavier an He initialization. Given those initializations the paper shows that one can trainer deeper models. Overall a nice paper with useful insights.